# Comprehensive Performance Evaluation of Green Infrastructure Practices for Urban Watersheds Using an Engineering–Environmental–Economic (3E) Model

**Yi-Jia Xing [1], Tse-Lun Chen [2]📷, Meng-Yao Gao [3], Si-Lu Pei [4], Wei-Bin Pan [1],\* and Pen-Chi Chiang [2],\***

[1] School of Environment and Energy, South China University of Technology, Guangzhou 510006, China; xing.yijia@mail.scut.edu.cn
[2] Graduate Institute of Environmental Engineering, National Taiwan University, 71 Chou-Shan Road, Da-an District, Taipei City 10673, Taiwan; d06541006@ntu.edu.tw
[3] Department of Chemical Engineering, National Taiwan University of Science and Technology, Taipei 10607, Taiwan; mygao@mail.ntust.edu.tw
[4] Research Institute of CNTY, Shanghai 200000, China; qzpsl92@outlook.com
\* Correspondence: ppwbpan@scut.edu.cn (W.-B.P.); pcchiang@ntu.edu.tw (P.-C.C.); Tel.: +86-13922205690 (W.-B.P.); +886-2-23622510 (P.-C.C.)

**Abstract:** Green infrastructure practices could provide innovative solutions for on-site stormwater management and runoff pollution control, which could relieve the stress of nonpoint pollution resulting from heavy rainfall events. In this study, the performance and cost-effectiveness of six green infrastructure practices, namely, green roofs, rain gardens, pervious surfaces, swales, detention basins, and constructed wetlands, were investigated. The comprehensive performance evaluation in terms of the engineering performance, environmental impact, and economic cost was determined in the proposed engineering–environmental–economic (3E) triangle model. The results revealed that these green infrastructure practices were effective for stormwater management in terms of runoff attenuation, peak flow reduction and delay, and pollutant attenuation. It was suggested that for pollution control, detention basins can efficiently reduce the total suspended solids, total nitrogen, total phosphorus, and lead. The implementation of detention basins is highly recommended due to their higher engineering performance and lower environmental impact and economic cost. A case study of a preliminary cost–benefit analysis of green infrastructure practice exemplified by the Pearl River Delta in China was addressed. It suggested that green infrastructure was cost-effective in stormwater management in this area, which would be helpful for sustaining healthy urban watersheds.

**Keywords:** urban watersheds; stormwater management; green infrastructure practices; multi-function; 3E triangle; strategies

## 1. Introduction

Healthy urban watersheds substantially affect the quality of life for people and the overall environment by providing many ecoservices. Naiman [1] suggested that a total of five components should be considered when identifying a healthy watershed: (i) basin geomorphology, (ii) hydraulic pattern, (iii) water quality, (iv) riparian characteristics, and (v) habitat characteristics. Conventionally, urban watersheds have been regulated by grey infrastructures, including dams, reservoirs, and channels, which can lead to impaired ecosystem quality and compromised hydraulic characteristics [2,3]. For this reason, the United States has restricted the massive construction of water conservancy infrastructures, prior to which, a comprehensive environmental impact evaluation should be performed [4]. Benedict and McMahon [5] suggested that green infrastructures (GIs) are superior to grey ones with respect to linkage, which regards a watershed area as a unit on a special and temporal scale. First, it has been widely acknowledged that GIs could efficiently reduce

combined sewer overflow events through enhanced infiltration [6]. The reduced runoff water can be stored in the infrastructure that is recharged back into the watershed during the dry seasons, maintaining the ecological base flow to sustain the ecoservice [7,8].

As an emerging structural stormwater control solution, green infrastructure has been adopted in many countries. For instance, the United States developed a list of the best practices to manage stormwater, sustain urban development, and ensure a healthy watershed [9]. The United Kingdom defined a sustainable urban drainage system to meet the requirement of draining capacity in heavy storm events [10]. Australia emphasized the importance of water issues in sustainable development and regarded stormwater control as a part of the water cycle in urban areas [11]. Based on these guidelines, a scientific approach to the development and implementation of green infrastructure could maximize the economic, social, and environmental benefits by achieving the highest engineering performance. For instance, it was pointed out that the implementation of green infrastructure would be based on ecological engineering principles, i.e., an integrated green infrastructure design incorporating multiple techniques could be more effective than that of single-design strategies [12].

Figure 1 illustrates 16 GI practices that are commonly used for stormwater management. According to their roles in stormwater control, the practices can be categorized into flow control, detention, infiltration, and treatment categories. Figure 1 depicts these categories in terms of the treatment volume and mechanism. Among these four types of GI practices, the capacity of runoff volume reduction and pollutant removal increases from flow control to treatment. Flow control includes soakwells, pervious surfaces, downspout disconnections, and green roofs, which are used for reducing the runoff volume at the source, such as rooftops and roads. Detention practices aim at providing temporary storage for runoff water and releasing it into the watershed at a controllable flow rate. When the runoff volume exceeds the capacity of detention practices, water infiltrating into the soil in the form of groundwater storage can prevent stormwater from directly entering sewer systems and water bodies. The last category is treatment, which includes proprietary treatment, constructed wetlands, swales, and living streams where the biological processes could play an important role in pollutant removal and water purification.

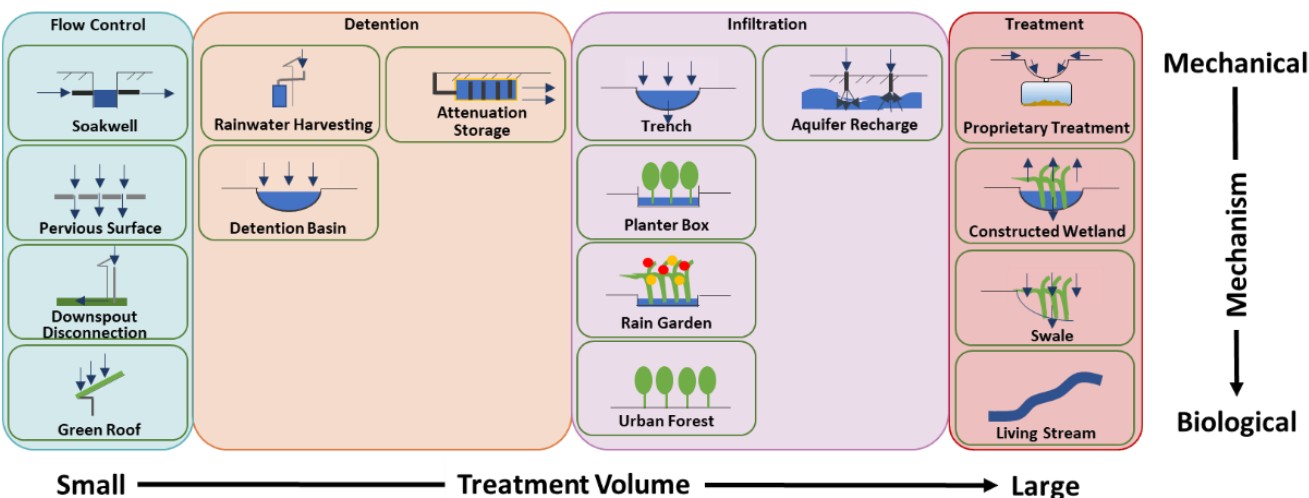

**Figure 1.** Recommended green infrastructure (GI) practices proposed by the United States, United Kingdom, and Australian authorities [10,11].

Life cycle assessment (LCA) provides an ad hoc approach to quantifying the environmental benefits of GIs. A previous study showed that a rain garden reduced the environmental impact by 62–98% while saving 42% in costs, making it an optimal option compared to grey infrastructure [13]. Flynn and Traver [14] found that the rain garden operation phase can reduce the amount of significant environmental impacts relative to

the construction phase impacts. Wang et al. [15] compared the environmental impacts of the combination of bioretention basins, green roofs, and permeable pavements with separate municipal stormwater sewer systems, and they found that the combination improved local water quality more cost-effectively by reducing the dependence on the local runoff quality. The above studies proved that GIs benefit the local water environment. However, little research has provided a solid selection matrix for choosing GIs to avoid environmental impact.

In addition, GIs can create a natural hydraulic process in an urban area, where the flow velocity can be significantly reduced and the hydraulic retention time can be drastically enlarged. Therefore, stormwater can be purified via biotic and abiotic processes. Moreover, the implementation of GIs in an urban area could increase biodiversity [16,17]. This would be beneficial to the self-purification capability of a watershed due to the intensified biotic, biochemical, and abiotic processes. Other ecoservices provided by GIs include carbon sequestration [18,19] and improving the air quality [20,21]. Moreover, numerous economic benefits, such as enhanced property values and sales can be achieved due to improved environmental quality [22]. For instance, Philadelphia Water Department planned to prevent combined sewer overflow events by implementing the green infrastructure practices in 2009 under the program named Green City, Clean Water [23,24]. An analysis showed the economic values of the program, with profit return for economic, social, and environmental benefits of 500 million USD, 1.3 billion USD, and 400 million USD respectively [25].

However, the lack of standardized approaches for the assessment of the performance of GI practices has made it difficult for engineers to select an appropriate practice [26–28]. For the development of the best available green infrastructure (BAGI), this study proposed a systematic approach for making urban watershed management plans. The objectives of this study were to (1) identify the available GI practices that are potentially applicable to urban watershed management, (2) establish the key performance indicators (KPIs) of GI practices for urban watershed management from the 3E (engineering, environmental, and economic) perspective, and (3) determine the BAGI practices for achieving urban watershed management.

## 2. Materials and Methods

### 2.1. Study Area

The study area was the Pearl River Delta (see Figure 2), which is the southeast coastal area of China, with a total area of 56,000 km². The population in this region was about 64.47 million in 2019, which was equivalent to ≈1049 capita/km². In addition, the gross domestic product in this area was about 13,766 billion USD in 2020 [29]. Meteorologically, the Pearl River Delta belongs to a subtropical monsoon climate zone, resulting in average annual precipitation of 1752.28 mm [30]. In addition, as it is located in a coastal area, the cities in the Pearl River Delta, such as Guangzhou, Shenzhen, Dongguan, and Foshan, are vulnerable to heavy rainfall events, resulting from severe climatic disasters, such as typhoons, which make it necessary to implement effective stormwater management strategies.

### 2.2. Framework for Determining the BAGI Practices

Figure 3 illustrates the systematically proposed framework for determining the BAGI practices. First, an overview of the available green infrastructure practices was performed. After that, the technical data of the different green infrastructure practices were gathered from the literature. Following Benedict and McMahon [31], our proposed framework included the importance of connectivity and the benefits of the ecosystem and community based on the ten principles of GI practices. The practices that met these criteria were thus designated as available GIs. In order to determine the BAGI practices, this study developed a 3E triangle model that was associated with a total of 14 KPIs, which provided a comprehensive assessment for the technologies from a standardized point of view using a graphical presentation [32,33]. The selection of different GI practices and the related KPIs

were identified by an expert consulting committee (i.e., an ad hoc committee) with the application of the Delphi method, which can be used to collect opinions and comments from different fields of expertise [29]. The KPIs were collected from the literature and screened in terms of data accessibility. In this study, 50%, 20%, and 30% of the invited experts were academics, industry professionals, and government officials, respectively. The Delphi study was conducted within two months, with the invited experts completing three rounds of online questionnaires. Based on the suggestions and/or comments from the committee, the integrated strategies for promoting GI implementation toward urban watershed management were proposed.

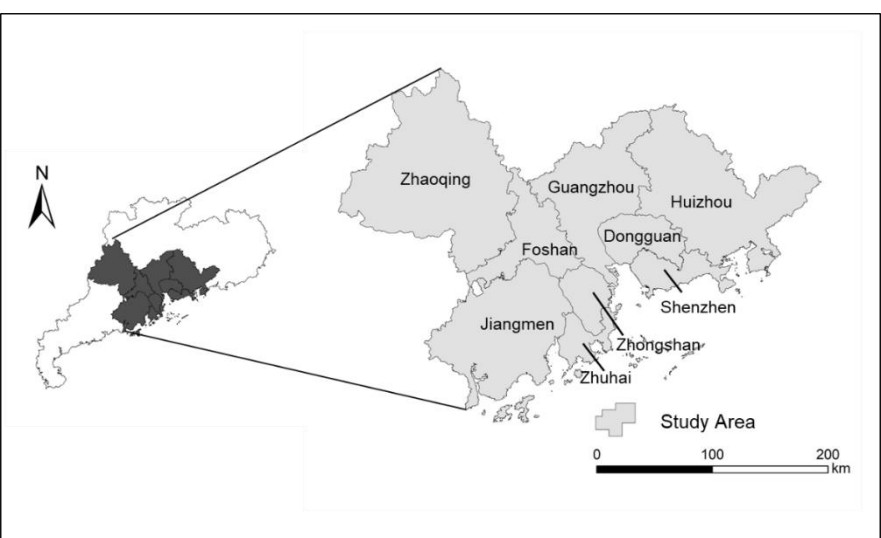

**Figure 2.** Location of the Pearl River Delta.

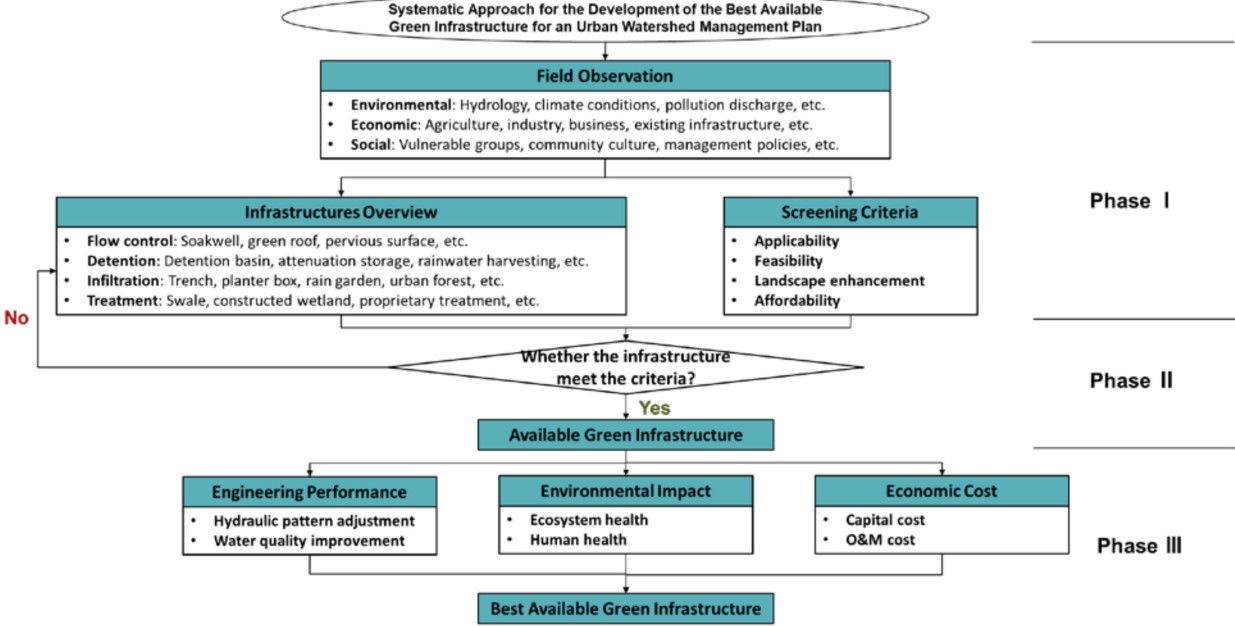

**Figure 3.** Framework used for the determination of the best available green infrastructure (BAGI) practices. O&M: operations and maintenance

Figure 4 illustrates the sequence of determining the BAGI practices using the Delphi method to select the available green infrastructure technologies through three phases. The selection criteria for determining the BAGI included the relationship with urban watershed

management, source control capability, availability of technical data, and functionality. Based on the technological maturity and the availability of technical data suggested by the ad hoc committee, the commonly used GI practices were preliminary narrowed down to a total of 11 candidates. The second round of selection involved the criteria regarding the effectiveness and feasibility. Pretreatment usually involves building a trench to maintain the appropriate functionality and prevent the clogging caused by coarse particles. A planter box is protected by vertical walls, which are higher than that of the grade level. Therefore, stormwater can flow into the box through the orifices in the wall; however, such a design limits the capability of water collection. The implementation of GI in an urban forest through multifunctional practices can effectively facilitate stormwater control, climate control, and air purification. However, the construction of an urban forest requires a large tract of land, which can limit the implementation in an urban area. Attenuation storage involves a centralized underground storage tank for collecting excessive runoff water. However, if implemented on its own, this practice negatively impacts the capability for water treatment. In addition, the construction and maintenance costs of attenuation storage facilities surpass those of surface systems. The proprietary treatment proposed by the United Kingdom is limited to specific types of land cover. Therefore, based on the aforementioned criteria, a total of six GI candidates, namely, green roofs, rain gardens, pervious surfaces, swales, detention basins, and constructed wetlands, were selected for the subsequent technology assessment via a 3E triangle model.

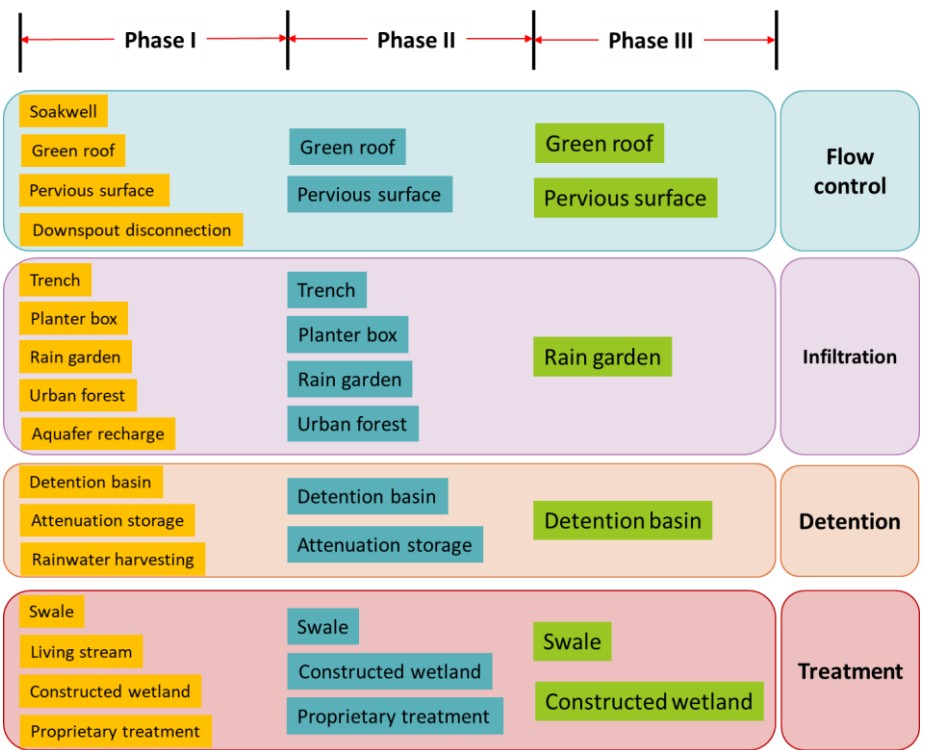

**Figure 4.** Selection procedure for determining the BAGI practices using the Delphi method.

### 2.3. Performance Evaluation of the Green Infrastructure Practices

A comprehensive performance evaluation of green infrastructures via the 3E triangle relies on a matrix of key performance indicators. Table 1 shows the 14 selected KPIs for evaluating the green infrastructure practices using the 3E model, which were selected by referring to our previous research [34] while considering the data accessibility. The engineering performance focused on the effectiveness of stormwater control, which involved the annual runoff attenuation, peak flow reduction, and peak flow delay. The engineering performance involved the peak flow delay and the reduction ratios of peak flow, total suspended solids (TSSs), total nitrogen (TN), total phosphorus (TP), and lead (Pb). The

quantitative evaluation of the environmental impact was inversely proportional to the pollutant abatement. Data on the effectiveness of the GIs regarding the pollutant reduction were derived from the International Stormwater Best Management Practice (BMP) database, which is maintained and updated by the United States Environmental Protection Agency (USEPA) [35]. The removal efficiency of different pollutants was calculated by comparing the median value of the concentration of the influent and effluent, which could be acquired from a box plot.

**Table 1.** The key performance indicators for evaluating the green infrastructure practices, along with their weighting factors, which were determined using the Delphi method.

| Aspects | Key Performance Indicators | | Units | $W_i$ | Remarks |
|---|---|---|---|---|---|
| Engineering performance (EP) | $EP_1$ | Peak flow reduction | % | 0.40 | Related to technological risk |
| | $EP_2$ | Peak flow delay | hours | 0.20 | Related to commercialization risk |
| | $EP_3$ | TSSs reduction | % | 0.10 | Related to technological risk |
| | $EP_4$ | TN reduction | % | 0.10 | Related to technological risk |
| | $EP_5$ | TP reduction | % | 0.10 | Related to technological risk |
| | $EP_6$ | Lead reduction | % | 0.10 | Related to technological risk |
| Life cycle environmental impact (LCEI) | $LCEI_1$ | Global warming potential | kg $CO_2$-Eq | 0.05 | Related to ecosystem risk |
| | $LCEI_2$ | Freshwater ecotoxicity | kg 1,4-DCB | 0.10 | Related to ecosystem risk |
| | $LCEI_3$ | Freshwater eutrophication | kg P-Eq | 0.20 | Related to ecosystem risk |
| | $LCEI_4$ | Human toxicity | kg 1,4-DCB | 0.35 | Related to human health risk |
| | $LCEI_5$ | Marine ecotoxicity | kg 1,4-DCB | 0.10 | Related to ecosystem risk |
| | $LCEI_6$ | Marine eutrophication | kg N-Eq | 0.20 | Related to ecosystem risk |
| Economic cost (EC) | $EC_1$ | Construction cost | USD/m$^2$ | 0.70 | Related to economic risk |
| | $EC_2$ | Operation and maintenance cost | USD/m$^2$ | 0.30 | Related to regulation risk |

$W_i$: weighting factor, TSSs: total suspended solids, TN: total nitrogen, TP: total phosphorus.

The LCA complied with the international standards of ISO 14040:2006 and ISO 14044:2006. The system boundary incorporated the construction phase (material extraction, energy production, and transportation) and operation phase. The environmental impacts of green infrastructure practices were calculated via the ReCiPe methodology using Umberto 5.6 software (ifu Hamburg GmbH, Hamburg, Germany) [36]. The results of LCA indicated the net environmental impact of the GIs, which represents the difference between the negative effects of the construction and maintenance and the positive effects of the stormwater management.

*2.4. Establishment of the 3E Triangle Model*

In this study, the 3E performances of the available green infrastructure practices were holistically assessed using a triangle model. Such 3E triangle models are popular for comprehensive performance evaluations [37,38], which can provide unique perspectives on engineering performance, environmental impact, and economic cost analysis. The 3E performance of each instrument directly resulted from the amounts of runoff attenuated and the pollutants intercepted during a certain period. As shown in our previous report [39], the triangle graphical presentation consists of the life cycle environmental impact (LCEI), engineering performance (EP), and economic cost (EC) on the *X-*, *Y-*, and *Z-*axes, respectively. Each axis was divided into five levels, i.e., very low, low, medium, high, and very high at intervals of 0.2. The areas within the triangular graph were divided into five zones, i.e., A (excellent), B (good), C (fair), D (poor), and E (worst). For example, points located in zone A are preferred because they provide very high performance with a very low environmental impact and a very low cost.

In this study, the 14 KPIs could be estimated as matrices for the LCEI, EP, and EC indicators, as shown in Equations (1) to (3), respectively:

$$\text{LCEI}_{yi} = (e_{yi}) = \begin{bmatrix} e_{11} & \cdots & e_{1n} \\ \vdots & \ddots & \vdots \\ e_{m1} & \cdots & e_{mn} \end{bmatrix}, \tag{1}$$

$$\text{EP}_{yj} = (p_{yj}) = \begin{bmatrix} p_{11} & \cdots & p_{1n} \\ \vdots & \ddots & \vdots \\ p_{m1} & \cdots & p_{mn} \end{bmatrix}, \tag{2}$$

$$\text{EC}_{yk} = (c_{yk}) = \begin{bmatrix} c_{11} & \cdots & c_{1n} \\ \vdots & \ddots & \vdots \\ c_{m1} & \cdots & c_{mn} \end{bmatrix}, \tag{3}$$

where $\text{LCEI}_{yi}$, $\text{EP}_{yj}$, and $\text{EC}_{yk}$ are the original matrices for the LCEI, EP, and EC indicators, respectively. "$y$" is the $y$th studied objects, which refer to the six different GIs; $i$, $j$, and $k$ refer to the $i$th selected LCEI indicator, the $j$th selected EP indicator, and the $k$th selected EC indicator, respectively.

Feature rescaling of the KPIs within the range of [0, 1] was adopted to make the features independent of each other due to the multiple dimension of KPIs and the variety of the value range of the KPI data. Two different feature scaling approaches were applied for two situations, which can be found elsewhere in our previous report [39]. Then, the synthetic $\text{KPI}_y$ indexes for LCEI, EP, and LCC could be calculated using Equation (4):

$$\text{KPI}_y = \sum_{i=1}^{m} (\text{KPI}_{yi} \times W_{yi}), \tag{4}$$

where $W_{yi}$ is the weighting factor of each KPI, which was determined by the ad hoc committee (expert consulting) using the Delphi method.

*2.5. Cost–Benefit Analysis of the Implementation of Green Infrastructure: A Case Study of the Pearl River Delta*

Li et al. [40] categorized the flooding with a depth of 20 cm to be a moderate flooding event. In the business-as-usual (BAU) scenario, they estimated that such a flooding event would affect 3.23 million people, 106,000 of them would be required to evacuate, and 60 of them would be killed. In addition, 8.98% of crop fields would be affected and 0.06% of houses would be devastated. Collectively, the flooding event at this level would result in an economic loss of 12.86 billion USD. The green infrastructure was a kind of on-site, decentralized stormwater management solution that could be implemented by combining the existing facilities.

A cost–benefit analysis was performed by assuming that the area was subjected to five moderate rainfall events of 20 cm events per year, which might cause 15-cm-depth flooding. The costs of green infrastructure implementation included the capital and maintenance costs, while the benefits included the avoidance of flooding-induced economic loss and stormwater treatment cost, accompanied by some environmentally related benefits, such as savings in electricity consumption and air quality improvement [41]. In this study, the costs and benefits were converted to the net present values (NPVs), as shown below:

$$\text{NPV} = \sum_{t=1}^{25} \frac{C_t}{(1+i)^t} - C_0, \tag{5}$$

where NPV (USD) is the net present value, $C_t$ (USD) is the net cash flow during the period $t$, $C_0$ (USD) is the capital investment, and the interest rate ($i$) was designated to be 8%, as suggested in the literature [42].

## 3. Results and Discussion

### 3.1. Engineering Performance of the Selected Available Green Infrastructure Practices

Figure 5 shows the effectiveness of selected GI solutions for pollutant removal. According to the analysis, green roofs turned out to be a source of TSSs, nitrogen, and phosphorous, such that they may not be suitable for pollution control. Except for green roofs, all other candidates efficiently reduced the TSSs from the stormwater runoff. Nitrogen in the runoff was found to be reduced by rain gardens [43], detention basins, and constructed wetlands. Such results might be ascribed to the intensified biological degradation and extended hydraulic retention time. The increase in nitrogen in the runoff from pervious surfaces results in dry deposition on the surface, making it difficult for green roofs, rain gardens, and swale systems to control the phosphorus due to the short hydraulic retention time [44]. Particularly, Davis and Stagge [45] suggested that the export of phosphorus is common in swales and could be attributed to the organic material in swales. All of the candidates exhibited more or less effectiveness regarding lead reduction, though a previous study suggested that the lead removed by pervious surfaces might be retained due to clogging of the particles [46].

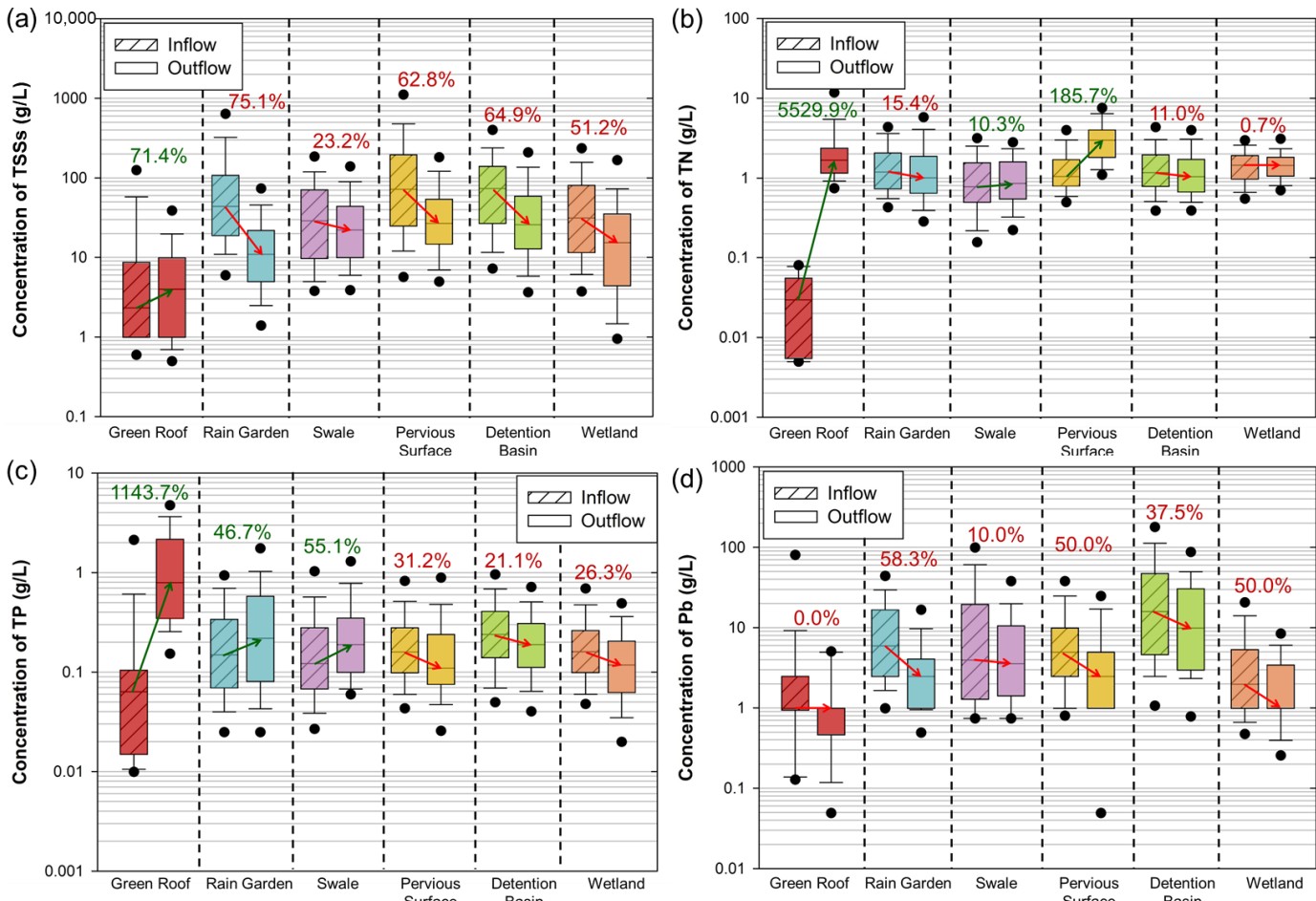

**Figure 5.** The effects of green infrastructure candidates on concentrations of (**a**) TSSs, (**b**) TN, (**c**) TP, and (**d**) Pb. The green numbers indicate increases in the pollutant concentrations in the effluent, while red ones indicate reductions.

Table 2 presents the pollutants removal mechanism of the GIs for stormwater purification exemplified by constructed wetlands. The solid particles are removed by sedimentation and filtration because the GI practices can extend the hydraulic retention time and establish flow barriers. The biological processes play an important role in nitrogen, phosphorus, metals, and organic matter removal. Ammonia ($NH_3$) can be easily dissolved into water and

then transferred into ammonium ions ($NH_4^+$) with neutralized pH values. Furthermore, biological nitrification and denitrification facilitate the serious transformation of $NH_4^+$ to nitrate ions ($NO_3^-$), nitrite ions ($NO_2^-$), and final elemental nitrogen forms. The major phosphorous treatments include adsorption (fixing phosphorous on the surface of solids), affecting the pH (adjusting the solubility and ionization behaviors of phosphorous), and mineralization (fixing phosphorous in a mineral matrix), which can control the phosphorous removal. The treatment of metal ions can be classified in terms of adsorption, cation exchange, microbial degradation, and plant uptake. The biotic and abiotic processes work together to facilitate the removal of metal ions. Therefore, the GIs serve as decentralized water treatment systems that not only improve the water quality but also eliminate non-point pollution. The implementation of GIs in urban areas could mitigate the rainwater flow rate and capture stormwater to supply the groundwater.

**Table 2.** Pollution treatment mechanism of green infrastructures exemplified by constructed wetlands.

| Pollutant | Mechanism | Description | Ref. |
|---|---|---|---|
| Solid | Sedimentation and filtration | GIs can provide extended hydraulic retention time, facilitating the sedimentation process. | - |
| Nitrogen | Volatilization | $NH_3(aq) + H_2O = NH_4^+ + OH^-$ | [47] |
|  | Nitrification | $NH_4^+ + 2O_2 = NO_3^- + 2H^+ + H_2O$ | [48] |
|  | Denitrification | $2NO_3^- \rightarrow 2NO_2^- \rightarrow 2NO \rightarrow N_2O \rightarrow N_2$ | [49] |
| Phosphorous | Adsorption and Reduction | The process is controlled by pH, redox potential, and mineral compositions. | [50] |
| Metals | Adsorption and cation exchange | Interaction with the soil matrix. | - |
|  | Microbial degradation | Metal could be oxidized in aerobic zones and/or be transformed to sulfides in anaerobic zones, both leading to facilitated precipitation. | [51] |
|  | Plant uptake | Soluble metals could be absorbed by plants and most of them accumulate in the roots. | [49] |

### 3.2. Life Cycle Assessment of the Selected Available Green Infrastructure Practices

Table 3 presents the life cycle inventory of six GIs used during construction, including the raw materials and processing. However, some natural materials, such as sand, gravel, and peat, might be excluded from the evaluation of environmental impacts during construction.

Figure 6 presents the environmental impacts of different GI practices in the form of endpoint scores. The environmental impacts resulting from material production, transportation, and the construction of GIs revealed that the pervious surface exhibited the most significant magnitude of the environmental impacts regarding all the selected impact categories. Such great impacts were attributed to the production of Portland cement. In addition, PVC manufacturing could also produce a great negative influence on the environment, which increased the impacts for the green roof, detention basin, and constructed wetland options because they used massive amounts of PVC as a liner to improve their impermeability. Rain gardens and swales were mainly comprised of naturally formed materials, i.e., sand, gravel, and peat, whose environmental impacts were $10^{-4}$ and $10^{-3}$ times that of the pervious surface. For the green roof, the environmental impact was also higher than the other four GI practices. This finding was in line with previous studies [55], which reported that the high impacts of a green roof were because of the utilization of a substrate and waste disposal. According to the engineering performance results, the high export of nitrogen and phosphorus may be the main reason for the risks to freshwater eutrophication, marine ecotoxicity, and marine eutrophication. These impacts are related to the maintenance of the green roof in terms of its fertilization.

**Table 3.** Life cycle inventory of six green infrastructures.

| Green Infrastructure Practices | | Construction | | References |
|---|---|---|---|---|
| | | Materials | Processing | |
| 1 | Green roof | Polyethylene: 0.46 kg/m² Polypropylene: 14.25 kg/m² PVC: 9.50 kg/m² | Material production | [52] |
| 2 | Rain garden | Sand: 58.58 kg/m² Clay: 37.59 kg/m² Gravel: 185.53 kg/m² | Material production Excavation: 0.26 m³ | [53] |
| 3 | Swale | Clay: 1079.18 kg/m² | Material production Excavation: 0.61 m³ | [53] |
| 4 | Pervious surface | Cement: 281.91 kg/m² Gravel: 281.91 kg/m² | Material production Excavation: 0.15 m³ | [53] |
| 5 | Detention basin | HDPE: 0.46 kg/m² Peat: 75.29 kg/m² Sand: 976.58 kg/m² PVC: 0.30 kg/m² | Material production Excavation: 1.22 m³ | [53] |
| 6 | Constructed wetland | Steel: 2.94 kg/m² PVC: 2.11 kg/m² Gravel: 719 kg/m² | Material production Excavation: 0.35 m³ | [54] |

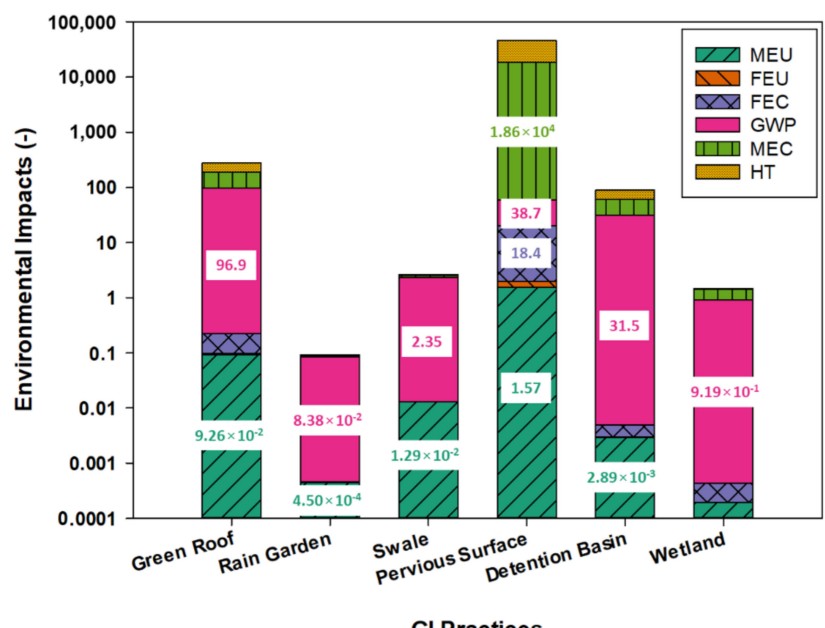

**Figure 6.** Endpoint assessment of different GI practices with values of significant impacts. MEU: marine eutrophication, FEU: freshwater eutrophication, FEC: freshwater ecotoxicity, GWP: global warming potential, MEC: marine ecotoxicity, HT: human toxicity.

In an extended life cycle, the implementation of green roofs, rain gardens, swales, and pervious surfaces might result in nutrient discharges that lead to eutrophication in the ecosystem. The construction of a 1 m² detention basin and constructed wetland would release $2.89 \times 10^{-3}$ and $1.95 \times 10^{-4}$ kg N-eq (indicated by marine eutrophication), accompanied by the release of phosphorous at $9.85 \times 10^{-5}$ and $4.64 \times 10^{-8}$ kg P-eq (indicated by freshwater eutrophication), respectively. In this case, such a negligible negative impact could be mitigated within a year. Therefore, we highly recommended that these recyclable and naturally formed construction materials be developed and implemented. In addition, as long as land availability is achievable, the implementation scale should be extended in such a way that the environmental impact can be diluted. In the end, deten-

tion basins and constructed wetlands have the potential for deployment in end-of-pipe stormwater control.

### 3.3. Evaluation of the Selected Best Available Green Infrastructure Practices of the 3E Model

According to Table 4, the detention capacity of a basin is usually much higher than the precipitation volume, though it cannot be assumed that the rainfall is completely retained. It was suggested that the implementation of a GI should consider the runoff volume and pollutants in the area. For instance, both the green roof and rain garden could be regarded as the decentralized stormwater control facilities for reducing the hydraulic and pollution loadings, respectively. The swale and pervious pavement are eligible for public areas to ensure safer traffic conditions. The effluent could be conducted into a detention basin and/or a constructed wetland for pollution abatement to reduce the loading of treatment facilities.

**Table 4.** Summary of engineering performance, environmental impact, and economic cost of green infrastructure practices.

| GI Instrument | | Green Roof | Rain Garden | Swale | Pervious Surface | Detention Basin | Wetland |
|---|---|---|---|---|---|---|---|
| Engineering Performance | Peak Flow Reduction (%) | 64.5 ± 21.4 [56–58] | 52.5 ± 14.8 [59,60] | 60.5 [61,62] | 86 [63] | 96.5 [64] | >80 [65] |
| | Peak Flow Delay (h) | 0.5 [56,66] | 1.5–3.0 [59,60] | 0.7 [67] | 1 [63,68] | 9.8 [69] | 48–72 [65,70] |
| | TSS Removal (%) | −71.4 | 75.1 | 62.8 | 23.2 | 64.9 | 51.2 |
| | TN Removal (%) | −5529.9 | 15.4 | −185.7 | −10.3 | 11.0 | 0.7 |
| | TP Removal (%) | −1143.7 | −46.7 | 31.2 | −55.1 | 21.1 | 26.3 |
| | Lead Removal (%) | 0.0 | 58.3 | 50.0 | 10.0 | 37.5 | 50.0 |
| Environmental Impact | GWP | 96.9 | $8.38 \times 10^{-2}$ | 2.35 | 38.7 | 31.5 | $9.19 \times 10^{-1}$ |
| | FEC | $1.27 \times 10^{-1}$ | $6.06 \times 10^{-6}$ | $6.97 \times 10^{-5}$ | 18.4 | $1.92 \times 10^{-3}$ | $2.46 \times 10^{-4}$ |
| | FEU | $3.72 \times 10^{-3}$ | $6.80 \times 10^{-9}$ | $1.95 \times 10^{-7}$ | $4.34 \times 10^{-1}$ | $9.85 \times 10^{-5}$ | $4.64 \times 10^{-8}$ |
| | HT | 88.6 | $2.70 \times 10^{-3}$ | $7.62 \times 10^{-2}$ | $2.71 \times 10^{4}$ | 28.0 | $4.48 \times 10^{-2}$ |
| | MEC | 92.3 | $5.37 \times 10^{-3}$ | $1.52 \times 10^{-1}$ | $1.86 \times 10^{4}$ | 29.2 | $5.23 \times 10^{-1}$ |
| | MEU | $9.26 \times 10^{-2}$ | $4.50 \times 10^{-4}$ | $1.29 \times 10^{-2}$ | 1.57 | $2.89 \times 10^{-3}$ | $1.95 \times 10^{-4}$ |
| Economic Cost | Capital Cost (USD/m$^2$) | 106 | 120 | 1.00 | 60.00 | 1.01 | 1.20 |
| | Maintenance Cost (USD/m$^2$) | 3.14 | 7.20 | 0.31 | 0.33 | 0.21 | 8.38 |

Figure 7 illustrates the results of the holistic assessment from the perspective of engineering performance, economic cost, and environmental impact, revealing that the detention basins and swales exhibited great capabilities regarding balancing these three aspects. The superiority of detention basins can be ascribed to their large detention capacity and extended hydraulic detention time, which could drastically relieve the hydraulic and pollution loadings in downstream areas. As for swale, the high ranking could be attributed to the relatively lower environmental impact, which was the result of the higher portion of natural materials, i.e., clays, used for its construction. In addition, the construction of detention basins is relatively easy compared to constructed wetlands, leading to higher cost-effectiveness. Compared with the environmental impact and economic cost, the engineering performance of swales was much more significant. Pervious surfaces impose great negative impacts on the environment due to the sophisticated material manufacturing process; however, these surfaces could be implemented in parking lots and roads to provide safer traffic conditions. As a result, more environmentally friendly material and processing technology are thus suggested. It should be borne in mind that the results only indicate the relative performances between the candidates. Therefore, though green roofs exhibited the worst performance and the highest environmental impact and economic cost, they could still be implemented for stormwater management in urban areas. As a result, more practical considerations should be put into the future implementation of KPIs.

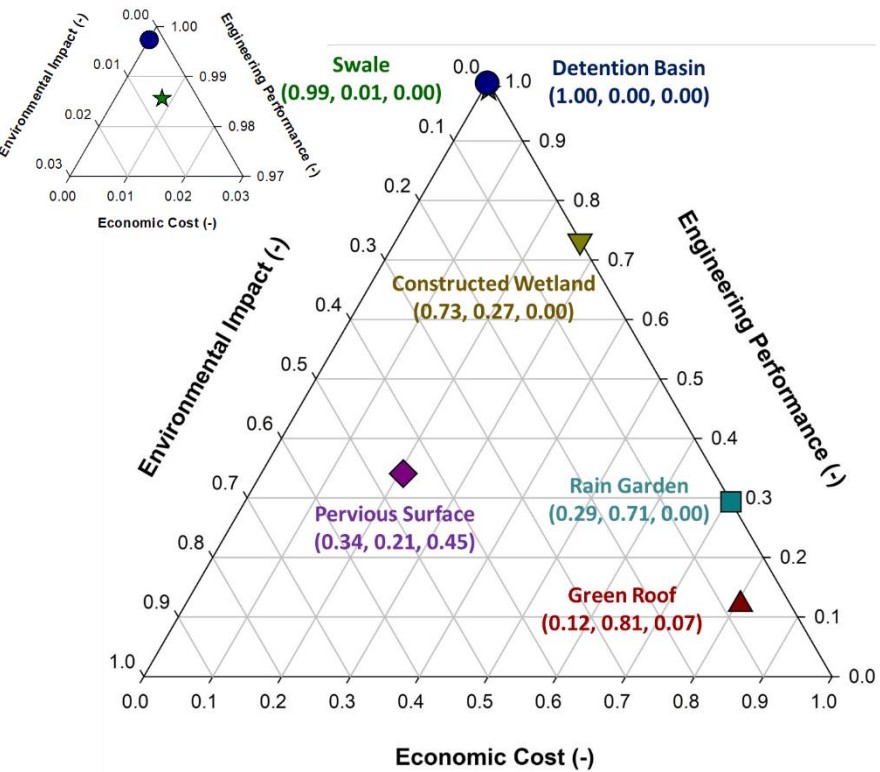

**Figure 7.** 3E triangle analysis of the best available green infrastructure practices. The numbers indicate the engineering performance, economic cost, and environmental impact, in that order.

Despite the growing enthusiasm for GI practices, some external issues need to be addressed. The most important issue is the limited land availability in urban areas. It has been shown that GIs do not work well in areas with low-permeability soils, steep slopes, or without enough space [71–73]. Even though GIs are effective in stormwater management, they must be installed on the surface of the land. As discussed above, wetlands and detention basins require more land to effectively collect the runoff water from urban areas. At this point, grey infrastructure can be successfully implemented underground, as the facility operations are supported by mechanical components. In addition, since GI practices are relatively new, many engineers and investors have less confidence in them [24]. Moreover, greater potential threats could emerge under intensified climate change conditions, which may exert more severe stress on GI in the foreseeable future [74].

*3.4. Preliminary Cost–Benefit Results of Implementation of Green Infrastructure in the Pearl River Delta*

A preliminary cost–benefit analysis of the implementation of GI practices in the Pearl River Delta was conducted and the result is shown in Figure 8. In the case study, swale and a detention basin had the highest rankings and were selected together with green roofs and pervious surfaces, which were effective regarding the runoff control at the source. Assuming the land cover of urban area in the Pearl River Delta was the same as the drainage area reported by Montalto et al. [75], the green infrastructure was fully implemented, and it could be estimated that the implementation area of green roofs, swale, pervious surfaces, and the detention basin were 9700, 6200, 4140, and 1750 km$^2$, respectively, it was expected that the runoff coefficient of the urban area would be reduced from 0.75 to about 0.39 [76]. It is noteworthy that the selected GIs for the case study were partly consistent with other research [76].

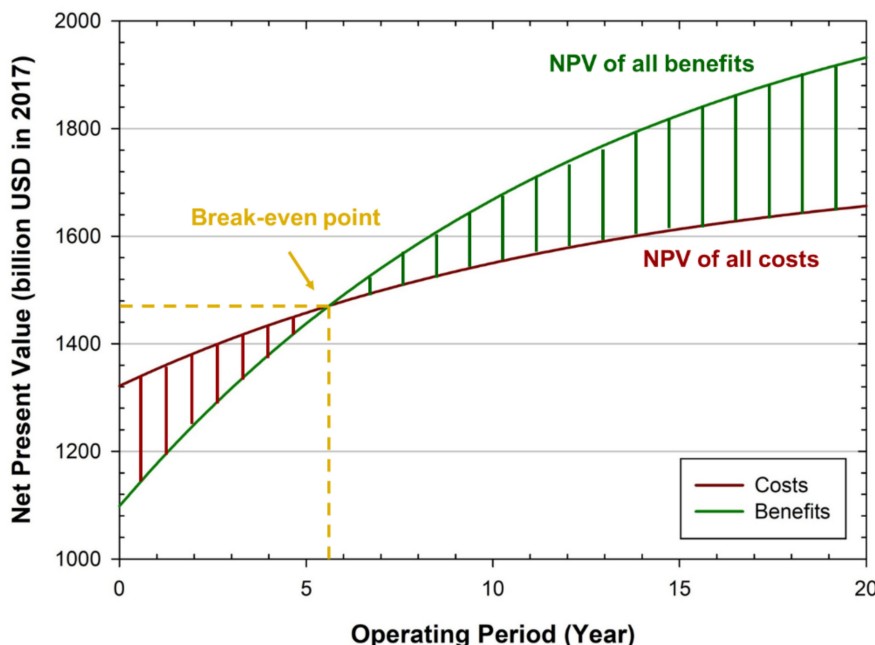

**Figure 8.** Break-even analysis of the net present value for determining the costs and benefits of green infrastructure (operating period: 20 years at an interest rate of 8%).

The capital cost of the green infrastructure would cost about 1287.85 billion USD which is equivalent to 99.9% of the GDP in this region in 2020, while the maintenance would cost about 34.04 billion USD each year. A direct benefit of green infrastructure implementation was the avoidance of a conventional stormwater treatment facility, whose capital cost was estimated to be 1088.35 billion USD, while the maintenance one was about 10.50 billion USD per year [77]. In addition, the green roof was effective regarding microclimate control, which was estimated to be able to save 3.40 billion USD of electricity consumption per year [78]. Furthermore, the air quality would be improved and about 0.02 billion USD of economic loss could be saved each year [78]. According to Figure 8, this suggested that the investment of green infrastructure could be retrieved in less than 6 years. After 20 years of implementation, the ratio of benefit to cost would be estimated to be 1.17. However, this quantified result could vary depending on the implementation scale of the BAGI.

## 4. Conclusions

This study proposed an innovative 3E method for screening and scoring six green infrastructure practices in terms of engineering, environmental, and economic performance. The method was exemplified by the urban watershed management in the Pearl River Delta.

Per the requirement of runoff regulations in the Pearl River Delta, six GIs, namely, green roofs, rain gardens, pervious surfaces, swales, detention basins, and constructed wetlands, were selected as the best available green infrastructure practices. According to the performance analysis, detention basins and constructed wetlands exhibited higher pollutant removal abilities, indicating the potential for serving as decentralized stormwater purification facilities. Wetlands can remove 51.2% TSSs, 0.7% nitrogen, 26.3% phosphorous, and 50% Pb from the inflow. Detention basins can remove 64.9% TSSs, 11.0% nitrogen, 21.1% phosphorous, and 37.5% Pb from the inflow. The high pollutant removal rate of wetlands and detention basins might result from the intensified biological degradation and extended hydraulic retention time.

The environmental impacts of the GIs were dependent on the construction materials to a large extent. In detail, rain gardens and swales presented a much lower impact than the other four GI practices. This was because rain gardens and swales were mainly comprised of naturally formed materials. Conversely, the pervious surfaces had high environmental impacts due to the consumption of Portland cement.

A comprehensive evaluation via a 3E triangle model revealed the superiority of detention basins and swale. Detention basins had better scores in terms of engineering performance, economic cost, and environmental impact (1.00, 0.00, 0.00). This was ascribed to their large detention capacity and extended hydraulic detention time, which could drastically release the hydraulic and pollution loadings in downstream areas. In addition, the construction of detention basins was relatively easy when compared with constructed wetland, leading to higher cost-effectiveness.

The case study of a full-scale implementation of green infrastructure in the Pearl River Delta indicated a promising result regarding the potential benefit of runoff control using GIs. Based on the cost–benefit analysis, it was concluded that the capital cost of the green infrastructure would cost about 1287.85 billion USD and the maintenance would cost about 34.04 billion USD each year. However, the investment of green infrastructure could be retrieved in less than 6 years. The ratio of benefits to costs was estimated to be 1.17 after 20 years of implementation. Overall, this study demonstrates the value of the 3E triangle model as a comprehensive performance evaluation of environmental, social, economic, and engineering performance toward selecting the best available green infrastructure practices.

To holistically develop green infrastructure, cross-disciplinary collaborations, including environmental, hydraulic, landscape, and civil engineering disciplines, should be established to optimize the engineering performance while maximizing the environmental, social, and economic benefits.

**Author Contributions:** Y.-J.X.: methodology, investigation, and writing—original draft preparation; T.-L.C.: methodology, writing—original draft preparation, and investigation; M.-Y.G.: conceptualization, writing—review and editing and methodology; S.-L.P.: methodology and formal analysis; P.-C.C. and W.-B.P.: project administration, supervision, and funding acquisition. All authors have read and agreed to the published version of the manuscript.

**Funding:** This research was funded by the Water Affairs Bureau of Yuexiu District, Guangzhou Municipality of the People's Republic of China under grant no. X2HJ-D5160090, Ministry of Science and Technology (MOST) of Taiwan under grant no. MOST 107-2221-E-002-009-MY3, and the Water Resources Agency of the Ministry of Economic Affairs (MOEA) of Taiwan under grant number MOEAWRA1070458.

**Institutional Review Board Statement:** Not applicable.

**Informed Consent Statement:** Not applicable.

**Data Availability Statement:** Not applicable.

**Acknowledgments:** Sincere appreciation goes to the Water Affairs Bureau of Yuexiu District, Guangzhou Municipality of the People's Republic of China, the Ministry of Science and Technology (MOST) of Taiwan, and the Water Resources Agency of the Ministry of Economic Affairs (MOEA) of Taiwan. P.-C.C. also wishes to thank Formosa Petrochemical Corporation for the professional suggestions regarding working on the green infrastructure practices on healthy urban watershed management.

**Conflicts of Interest:** The authors declare no conflict of interest.

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
