# Peer review of "Comprehensive Performance Evaluation of Green Infrastructure Practices for Urban Watersheds Using an Engineering–Environmental–Economic (3E) Model"

_sustainability, doi:10.3390/su13094678_

Round 1

Reviewer 1 Report

The paper is very interesting. It is also important from the point of view of water management in urbanized areas. Green Infrastructure practices are becoming increasingly more popular and in the article, their performance and cost effectiveness were investigated. The paper is based on a wide range of current literature. My minor comments are given down below:

  1. You give a short description of the classification and roles of common GI practices in the Results (in 3.1. section), but it would be nice if you can give some background to the GI practices - the categories, types, and short description of each of them.
  2. Continuing the previous comment, in section 3.1. you mention the selection of available GI practices. However, in section Materials and Methods, you provide the effects of their selecting and give the classification in Figure 2. In my opinion, the first paragraph of section 3.1. could be moved to the Introduction, as well as Figure 4. 
  3. Figure 2 is clearly presented. It would be good however to complete the selection criteria next to phases.

Reviewer 2 Report

Comprehensive Performance Evaluation of Green Infrastructure Practices for Urban Watersheds using an Engineering-Environmental-Economic (3E) Model

Overall: The theme of the paper is very relevant and interesting, and it really seems to be very innovative. However, the paper has some content problems. Some sections are not well explained and detailed, and some incongruencies appear in the text. The English also needs serious improvements. In order to be publishable, the paper needs major revisions.

Abstract: The abstract is ok, however it could be better explained. Its somehow confusing. The authors also state that the case study of this research was a pearl river delta in China, however the case study is never mentioned in the text ever again. Please enlighten me.

Introduction: The introduction is a little confusing. There is no flow in the ideas that the authors want to expose. Everything seems disconnected and with little explanations.it needs improvements. However, the objective of the paper is well explained.

Materials and Methods: This section is well explained however I feel it could be better written and more detailed. Some parts of the methodology are a bit confusing to the reader, and I would suggest to clarify and simplify the different steps you took in your research, especially in subsection 2.1. When it comes to section 2.2, it is not clear how do you come up with the Key Performance Indicators in table 1. They were a result of the Delphi method? Or were they previously chosen, and the Delphi method only gave you the weighting factors of the KPI?? It is not clear.

Results: This section is confusing. For example, in subsection 3.1, you present in figure 4 the 16 green infrastructure practices proposed by USA, UK and Australia, however in figure 2, you had already selected the 6 you were going to work with…why show again the 16 GI practices and not the final 6?? Very confusing. What is the point of table 2 if you only show the Pollution treatment mechanism of only one GI practice? What is its connection with the results? Plus, I don’t see a discussion section in this paper, so I am assuming this section is both the results and the Discussion. So, I find the discussion of the results very incomplete. A lot more can be said from your results. You should present results from other similar studies and find differences or similarities between them. Other thing that is really confusing is…how did you test these results? What was your case study? You referred in the abstract however you did not mention anything about that in the text. If there is a case study, please present it in the methods section, for example.

Conclusions: The conclusions are ok, however they could be more elaborate.

Round 2

Reviewer 2 Report

2nd Round Review

Comprehensive Performance Evaluation of Green Infrastructure Practices for Urban Watersheds using an Engineering-Environmental-Economic (3E) Model

Overall: The paper has improved a lot, however there are some more issues that should be improved. However, some minor issues must be addressed to be publishable, which are described in each section comments below. Nevertheless, I still feel the case study is not properly described. Please consider add a subsection in methods or in the introduction to introduce the case study (add a map and a descriptive summary for example). Additionally, the English has improved a lot, but it still needs some revisions.

Abstract: nothing to add

Introduction: This section has improved a lot. However, the paragraph related with the pearl river delta seems disconnected with the rest of the text. Consider remove it or rewrite it in order to flow with the next paragraph. I would suggest to remove this paragraph and place it in a subsection in Methods with the study area characterization.

Methods: This section is ok. However, it is still not clear to me how you came up with the KPI. Maybe you should add the sentence you use in the previous response, that is: The KPIs were collected from literatures and screened by data accessibility. It is much easier to understand. Another issue with this section is section 2.4. Although I understand the cost benefit analysis, why did you added “Case study of Guangzhou” when you did not mention this study on the text below? Please explain.

Results: Results are much better presented and discussed. Nothing to add

Conclusions: ok
